# Enhancement by Metallic Tube Filling of the Mechanical Properties of Electromagnetic Wave Absorbent Polymethacrylimide Foam

**DOI:** 10.3390/polym11020372

**Published:** 2019-02-20

**Authors:** Leilei Yan, Wei Jiang, Chun Zhang, Yunwei Zhang, Zhiheng He, Keyu Zhu, Niu Chen, Wanbo Zhang, Bin Han, Xitao Zheng

**Affiliations:** 1School of Aeronautics, Northwestern Polytechnical University, Xi’an 710072, China; yanleilei@nwpu.edu.cn (L.Y.); c.zhang@nwpu.edu.cn (C.Z.); hzh633@mail.nwpu.edu.cn (Z.H.); zhukeyu@mail.nwpu.edu.cn (K.Z.); 1120734419@mail.nwpu.edu.cn (N.C.); zhengxt@nwpu.edu.cn (X.Z.); 2School of Mechanical Engineering, Xi’an Jiaotong University, Xi’an 710049, China; 3Department of Basic Sciences, Air Force Engineering University, Xi’an 710051, China; jwow918@163.com (W.J.); zhang_yunwei@126.com (Y.Z.); m13201689787@163.com (W.Z.); 4Research Institute of Xi’an Jiaotong University, Hangzhou 311215, Zhejiang, China

**Keywords:** absorbent PMI foam, metallic tube, electromagnetic wave absorption, mechanical properties, failure mechanism

## Abstract

By the addition of a carbon-based electromagnetic absorbing agent during the foaming process, a novel electromagnetic absorbent polymethacrylimide (PMI) foam was obtained. The proposed foam exhibits excellent electromagnetic wave-absorbing properties, with absorptivity exceeding 85% at a large frequency range of 4.9–18 GHz. However, its poor mechanical properties would limit its application in load-carrying structures. In the present study, a novel enhancement approach is proposed by inserting metallic tubes into pre-perforated holes of PMI foam blocks. The mechanical properties of the tube-enhanced PMI foams were studied experimentally under compressive loading conditions. The elastic modulus, compressive strength, energy absorption per unit volume, and energy absorption per unit mass were increased by 127.9%, 133.8%, 54.2%, and 46.4%, respectively, by the metallic tube filling, and the density increased only by 5.3%. The failure mechanism of the foams was also explored. We found that the weaker interfaces between the foam and the electromagnetic absorbing agent induced crack initiation and subsequent collapses, which destroyed the structural integrity. The excellent mechanical and electromagnetic absorbing properties make the novel structure much more competitive in electromagnetic wave stealth applications, while acting simultaneously as load-carrying structures.

## 1. Introduction

Multi-functional designs of materials and structures, such as collaborative design of mechanical and electromagnetic (EM) wave absorption, which is critical in aircraft and aerospace applications, are more and more attractive and have been widely studied. Metamaterial absorber (MMA) is a kind of composite material, which usually consists of periodic artificial structures and dielectric substrates [1,2]. By transforming the electromagnetic wave energy into other forms (e.g., as thermal energy), MMA can exhibit electromagnetic wave absorption [3,4,5]. A frequency-selective surface (FSS) absorber consists of lossy resistive patches, and the performance of broadband absorbing can be easily obtained by reasonable design of the planar patterns [6,7,8]. Three-dimensional structures [9,10,11] were also developed for wide-band and wide-angle electromagnetic wave absorption, such as folded resistive patches [9] and honeycombs [12]. Such structures have excellent electromagnetic (EM) wave absorption properties, but researchers always fail to consider their mechanical properties. Besides metamaterial, the electromagnetic properties of carbon foam [13], graphene foam [14], CNT (carbon nanotube) and graphene composites [15,16], polyurethane foam [17], and other polymer-based materials [18] have also been considered, but they usually have poor mechanical performances.

Polyimide foams act as lightweight structure materials and have outstanding strength-to-weight ratios, specific stiffness, and specific energy absorption (SEA) [19,20,21,22]; among these are polymethacrylimide (PMI) and polyetherimide (PEI) foams. Their mechanical properties have been widely studied, such as elastic [19] and viscoelastic behavior [20], dynamic resistance [21], and fatigue [22], as well as the mechanical performances of PMI foam-cored sandwich structures [22,23]. However, with relative permittivity of 1.06, such PMI foam does not contribute to electromagnetic wave absorption [24,25]. Adding an electromagnetic wave-absorbing agent during the foaming process may enhance the electromagnetic wave absorption properties, but the agent addition also generates surfaces, which may cause a decrease of mechanical performance.

Because of the respective limitation of the above metamaterial absorber (MMA), frequency-selective surface (FSS), and PMI foam structures, efforts have been made to form novel materials and structures with combined mechanical and electromagnetic properties in recent years. Structures with honeycombs [26,27,28], hierarchical lattice [29], and local stitched radar-absorbing structures [30] have been developed to have both specific strength and electromagnetic wave absorption properties. Besides structure design, it has also been demonstrated that tubes could increase the mechanical performances of foam structures effectively, such as PMI foam [31,32] and aluminum foam [33].

By a filling of metallic tube and the addition of electromagnetic absorbing agent into a PMI foam, it can be expected to obtain a novel cellular structure exhibiting excellent mechanical and electromagnetic wave absorption properties. In the present study, carbon-based electromagnetic absorbing agent is added to a PMI foam during the foaming process, and forms a novel absorbent PMI foam. Moreover, metallic tubes are employed as a filling material to enhance the mechanical properties of the absorbent PMI foam. Their mechanical behaviors are studied experimentally under compressive loading. The enhancement of elastic modulus, compressive strength, and specific energy absorption (SEA) of both normal and absorbent PMI foams by metallic tube filling is discussed, and the failure mechanism is explored.

## 2. Experimental Measurement

### 2.1. Materials and Fabrication

Commercial polymethacrylimide (PMI) foam was employed in the present study. By filling with a carbon-based electromagnetic absorbing agent during the foaming process, a novel electromagnetic wave absorbent PMI foam could be obtained (samples fabricated by Hunan Zihard Material Technology Co. Ltd, Hunan, China). The density of the absorbent PMI foam was 222 kg/m^3^, increased only by 5.6%, compared to that of the normal PMI foam (210.3 kg/m^3^). Metallic circular tubes, made of 6061 aluminum alloy and 304 stainless steel, were chosen as the fillers to enhance the normal and absorbent PMI foam. The outer diameters of the metallic tubes were fixed to *Φ* = 10 mm, and the wall thicknesses were 0.5 and 0.2 mm for aluminum alloy and stainless steel, respectively.

The PMI foams were firstly cut into square cubes with the dimensions of 40 mm × 40 mm × 40 mm and then perforated to form a through hole in the middle region. The metallic tubes were cut by electro-discharge machining (EDM) to fit into the prepared holes in the foam block. PMI cylinders were also prepared to be inserted into the inner interspace of the metallic tubes, to form the foam-filled tube-enhanced PMI foam. The metallic tube, PMI foam matrix, and foam cylinder were assembled and subsequently stuck together by epoxy glue to get the tube-enhanced PMI foam. Typical specimens of normal and absorbent PMI foams, tube-enhanced PMI foams, and foam-filled tube-enhanced absorbent PMI foams are shown in Figure 1. It is noted that the averaged density $\rho_{c}$ is defined as the total mass of specimen divided by the whole volume (40 mm × 40 mm × 40 mm).

### 2.2. Measurement of Electromagnetic Wave Absorbtion

The electromagnetic wave absorption of the absorbent PMI foam was characterized by the measurement of reflection. As shown in Figure 2a, the measurement was carried out at the frequency of 2–18 GHz by using the free-space method in a microwave anechoic chamber at ambient temperature, through the test system based on an Agilent E8363B Network Analyzer. The electromagnetic wave reflection property of normal and absorbent PMI foams with the dimensions of 600 mm × 600 mm × 20 mm were measured. A metal plate with the same size of the specimen was employed as the back plate to avoid wave transmission. It is noted that the reflection from a metal plate used as a prototype should be firstly measured, for the sake of normalization.

### 2.3. Compressive Tests

The detailed parameters of normal and absorbent PMI foam specimens for compressive tests are listed in Table 1. Quasi-static compression was carried out by an electronic universal testing machine (INSTRON-3382) at room temperature. The loading rate was fixed at 2 mm/min, with a nominal strain rate less than 10^−3^ s^−1^. The compressive strain of at least 75% was achieved for each specimen to ensure complete deformation and energy absorption. Digital images of each specimen were acquired to capture the deformation modes and explore the failure mechanisms. No less than three specimens in each case were measured in the tests to acquire the averaged mechanical properties.

## 3. Results and Discussion

### 3.1. Electromagnetic Wave Absorbtion

The electromagnetic wave absorptivity can be expressed as [11,26]:(1)$$A=1-|S_{11}{|}^{2}-|S_{21}{|}^{2}$$
where A represents the electromagnetic wave absorptivity, while $|S_{11}{|}^{2}$ and $|S_{21}{|}^{2}$ represent the reflectivity and the transmissivity, respectively. Note that, in this calculation, $S_{11}$ and $S_{21}$ are linear values. Here, $|S_{21}{|}^{2}$ is equal to 0, due to the employment of the metal backboard in the present measurement.

Figure 2b shows the reflectivity of the vertical incident waves ($S_{11}$) at the frequency of 2–18 GHz. The average reflectivity of the absorbent PMI foam at the frequency of 4.9–18 GHz was less than −8 dB, which implies that the electromagnetic wave absorptivity A, calculated by Equation (1), can be larger than 85%. At the frequency of 5.2–7.3 GHz, 9.9–12.85 GHz, and 14.5–18 GHz, the reflectivity was even less than −10 dB, with the absorptivity larger than 90%. At the specific range of 6.25–6.55 GHz, the reflectivity decreased to less than −15 dB. In contrast, the electromagnetic waves were completely reflected by the normal PMI foam (see Figure 2b). This implies that the present absorbent PMI foam possessed good electromagnetic wave-absorbing properties.

### 3.2. Compressive Strength and Energy Absorption

#### 3.2.1. Enhancement of Compressive Strength

Figure 3a compares the compressive stress versus strain curves of normal and absorbent PMI foams. All specimens exhibited foam-like features, i.e., the three typical regions including linear, plateau, and densification regions [34]. The absorbent PMI foam had similar density to the normal one, but the compressive strength $\sigma_{peak}$ (the first peak stress) and plateau strength were even less than half of those of the normal foam. It can be seen from Figure 4 that during the compressive process, cracks could be obviously observed in the absorbent PMI foam specimen, which caused subsequent collapses, while for the normal foam, only compaction occurred leading to densification.

A metallic tube was employed as a filler to improve the poor compressive performance of the absorbent PMI foam. Figure 5a and Figure 6a present the typical stress-versus-strain curves of the tube-enhanced normal and absorbent PMI foams, respectively. It is shown that filling of both aluminum and 304 stainless steel tubes led to significant enhancement of the compressive performances of both normal and absorbent PMI foams. The tube-enhanced PMI foams still underwent foam-like features but exhibited some obvious fluctuations in the plateau region. The jagged stress-versus-strain curves presented multiple peaks and valleys before densification. Each stress peak and valley were related to one folding (i.e., progressive buckling deformation [23,35]), which could be demonstrated through the buckling deformation mode layer by layer, as shown in Figure 7d. Moreover, internal filling of the PMI foam into 304 stainless steel tube (specimen 9) led to a significant increase of the compressive strength of the tube-enhanced absorbent PMI foam, while no obvious improvement could be seen in the case of the aluminum tube-enhanced foam (specimen 7). This may be attributed to the fact that the wall thickness of the 304 stainless steel tube (0.2 mm) was smaller than that of the aluminum tube (0.5 mm), and the buckling mode of the tube with the thinner wall could be more easily affected by the internal filling of foam. Therefore, the internal filling of foam changed the buckling mode of the 304 stainless steel tube and led to the increase of compressive strength.

The averaged elastic modulus E and the compressive strength $\sigma_{\mathrm{peak}}$ are summarized in Table 2. For the absorbent PMI foam, filling with the metallic tube only contributed a little to the increase of density (less than 5.3%) but increased the elastic modulus E of the normal PMI foam by 77.7% (specimen 3), and that of the absorbent PMI foam by 128% (specimen 9). Moreover, the peak compressive strength $\sigma_{\mathrm{peak}}$ was increased by 18.3% (specimen 4) for the normal PMI foam, and surprisingly by 133.8% (specimen 9) for the absorbent PMI foam.

#### 3.2.2. Enhancement of Energy Absorption

The energy absorption capacity is commonly characterized by energy absorption per unit volume $W_{v}$:(2)$$W_{V}=\int_{0}^{\bar{\varepsilon }}\sigma d\varepsilon$$

The energy absorption per unit volume $W_{v}$ as a function of compressive strain of the PMI foam, tube-enhanced PMI foam, and tube-enhanced absorbent PMI foam, is shown in Figure 3, Figure 5b, and Figure 6, respectively, and the calculated $W_{v}$ at $\bar{\varepsilon }$ = 0.5 are summarized in Table 2. As shown in Figure 3b, the absorbent PMI foams were not as effective as the normal ones in energy absorption, for the electromagnetic absorbing agent addition in PMI foam caused an increase of brittle features, which decreased the compressive strength. It was found that the tube filling influenced the energy absorption of both normal and absorbent PMI foams.

In addition, the specific energy absorption (SEA) was another important parameter in weight-sensitive applications, which could be defined as averaged energy absorption per unit mass [34]:
(3)$$W_{m}=\frac{W_{v}}{p_{c}}$$

The $W_{m}$ of the specimens are also shown in Table 2. For normal PMI foams (specimens 1–4), the energy absorption per unit volume $W_{v}$ and per unit mass $W_{m}$ by filling the metallic tube was increased by 18% (specimens 3 and 4) and 6.7% (specimen 4), respectively. A much more obvious enhancement was found for the tube-enhanced absorbent PMI foams (specimens 5–9): $W_{v}$ and $W_{m}$ of tube-enhanced absorbent PMI foams were increased by 54.2% (specimen 9) and 46.4% (specimen 9), respectively.

#### 3.2.3. Failure Mechanism

Figure 7 presents the specimen images of the normal PMI foam, tube enhanced, and foam-filled tube-enhanced PMI foams after compression. For all three kinds of specimens, the structural integrity could be well ensured. During the compressive process, the normal PMI foam underwent compaction and densification, showing ductile collapse. In contrast, the absorbent PMI foam exhibited brittle collapse, with cracking and delamination occurring due to the addition of the electromagnetic absorbing agent. As shown in Figure 8, the specimens were all damaged completely after compression (with compressive strain over 75%) and broken into small pieces of different sizes. In local fractography in Figure 8d, the interfaces between the foam and the absorbing agent of the absorbent PMI foam can be seen clearly. The poor interfaces with weak bonding strength were more prone to initiate a crack, leading to delamination and subsequent collapse of the specimen. Figure 8e,f show the fracture surfaces of the absorbent PMI foam with different magnifications.

Adding the absorbing agent improved the electromagnetic wave absorption of the absorbent PMI foam. Meanwhile, the weaker interfaces between the foam and the absorbing agent also decreased its mechanical properties. Therefore, the present effective enhancement approach by metallic tube filling is of great significance for engineering applications.

## 4. Comparison

The specific compressive strength $\sigma_{peak}/\rho_{c}\sigma_{Y}$ (here, $\sigma_{Y}$ refers to the yielding strength of the metallic tubes) and energy absorption per unit mass $W_{m}$ (SEA) of the present tube-enhanced normal and absorbent PMI foams were compared with those of other competing metallic sandwich cores [36]. As shown in Figure 9, the present tube-enhanced PMI foams were quite competitive especially in energy absorption. Compared with other metallic lattice cores, which have shown significant advantages in load-carrying applications [37,38], the present tube-enhanced PMI foam seemed more effective. By optimizing the design of the geometric parameters in the future, the present tube-reinforced structure will be more competitive.


## 5. Conclusions

In conclusion, a PMI foam was endowed with the property of electromagnetic wave absorption by adding an electromagnetic absorbing agent during the foaming process, to form a novel absorbent PMI foam. Metallic circular tubes, made of 6061 aluminum alloy and 304 stainless steel, were chosen as the fillers to enhance the mechanical performance of the normal and absorbent PMI foams. The properties of electromagnetic wave absorption, as well as compressive strength and energy absorption, were experimentally investigated. The main findings are summarized as follows:

(1) The absorbent PMI foam exhibited good electromagnetic wave absorption, with electromagnetic wave absorptivity larger than 85% at a large frequency range of 4.9–18 GHz. The absorptivity even exceeded 90% at a specific range of frequency.

(2) During the compressive process, the normal PMI foam underwent compaction and densification, showing ductile collapse. In contrast, the absorbent PMI foam exhibited brittle collapse, with cracking and delamination occurring due to the addition of the electromagnetic absorbing agent.

(3) A filling of metallic tubes increased the mechanical properties of both normal and absorbent PMI foams, and the enhancement was greater for the absorbent PMI foam. The elastic modulus E, compressive strength $\sigma_{\mathrm{peak}}$, and energy absorption per unit volume $W_{v}$ and per unit mass $W_{m}$ of the tube-enhanced absorbent PMI foam could be increased by 127.9%, 133.8%, 54.2%, and 46.4%, respectively, with the density increasing only by 5.3%. Filling with a 304 stainless steel tube was more effective than filling with an aluminum tube.

(4) With their outstanding performances in electromagnetic wave absorption, compressive strength, and energy absorption, the proposed tube-enhanced absorbent PMI foam is quite competitive in applications such as simultaneous electromagnetic wave stealth, load carrying, and impact resistance.

## Figures and Tables

**Figure 1 polymers-11-00372-f001:**
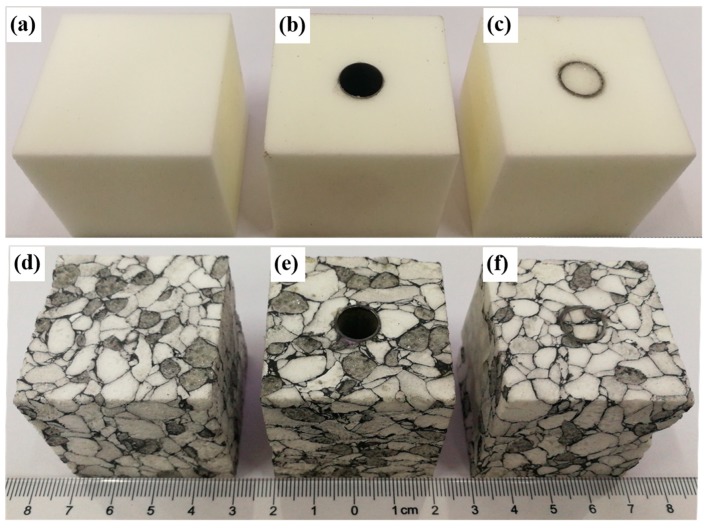
Specimen images of polymethacrylimide (PMI) foam. (**a**) PMI foam; (**b**) Tube-enhanced PMI foam; (**c**) Foam-filled tube-enhanced PMI foam; (**d**) Absorbent PMI foam; (**e**) Tube-enhanced absorbent PMI foam; (**f**) Foam-filled tube-enhanced absorbent PMI foam.

**Figure 2 polymers-11-00372-f002:**
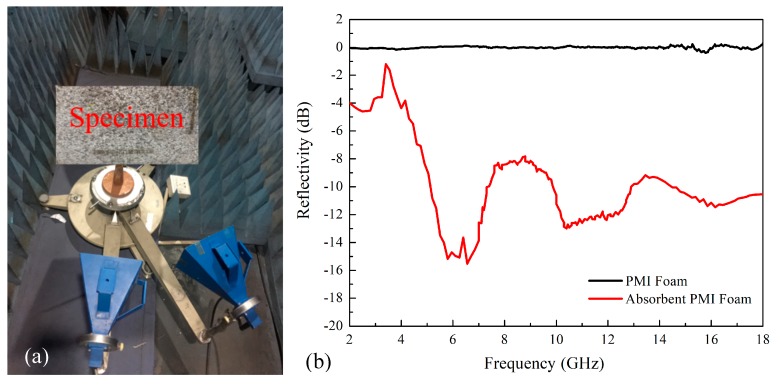
The wave-absorbing properties of normal and absorbent PMI foams; (**a**) Experimental setup; (**b**) Experimental results of reflectivity for vertical incident waves.

**Figure 3 polymers-11-00372-f003:**
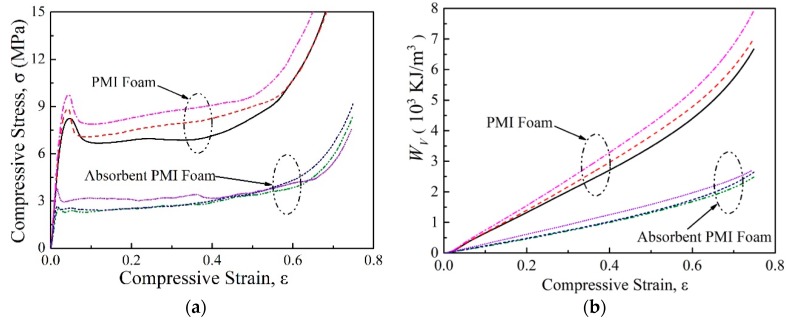
Compressive behaviors of PMI foams. (**a**) Stress strain curve; (**b**) Energy absorption. The ellipse circles indicate classes of specimens.

**Figure 4 polymers-11-00372-f004:**
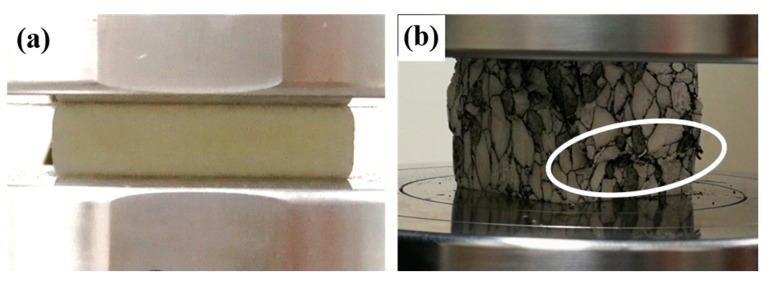
Images of PMI foams during compression. (**a**) PMI foam at compressive strain of 75%; (**b**) Absorbent PMI foam at compressive strain of 30%. The ellipse region shows where cracks occurred which subsequently caused collapse.

**Figure 5 polymers-11-00372-f005:**
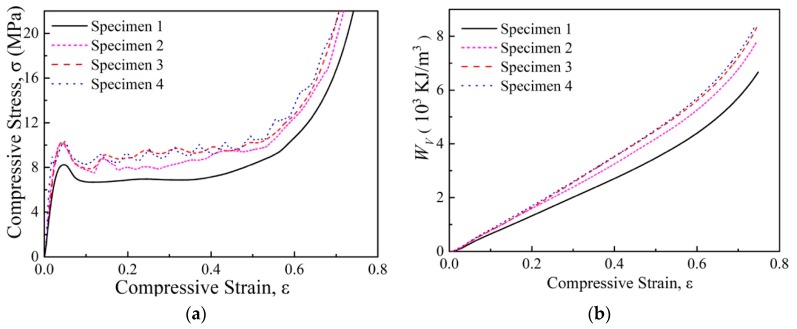
Compressive behaviors of tube-enhanced PMI foams. (**a**) Typical stress strain curves; (**b**) Energy absorption per volume.

**Figure 6 polymers-11-00372-f006:**
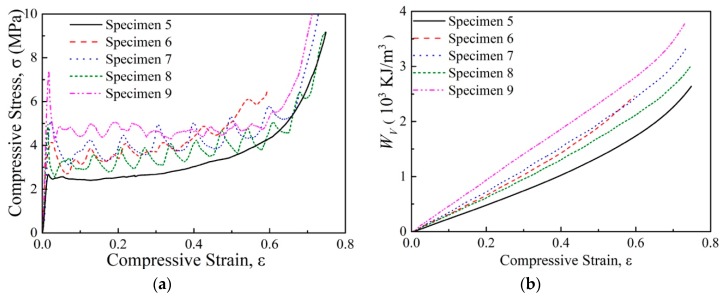
Compressive behaviors of tube-enhanced absorbent PMI foams. (**a**) Typical stress–strain curves; (**b**) Energy absorption per volume.

**Figure 7 polymers-11-00372-f007:**
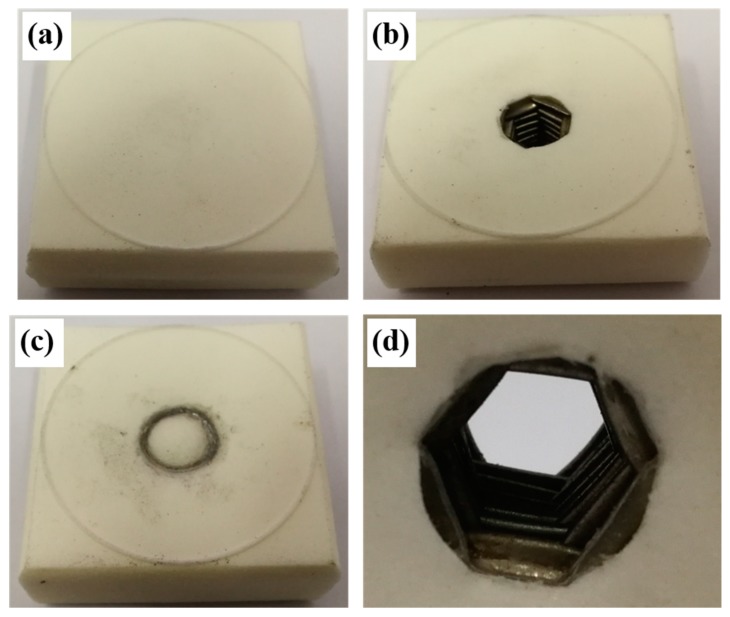
Normal PMI foam-based specimen images after compression. (**a**) PMI foam; (**b**) Tube-enhanced PMI foam; (**c**) Foam-filled tube-enhanced PMI foam; (**d**) Layer-by-layer buckling deformation of metallic tube in tube-enhanced PMI foam.

**Figure 8 polymers-11-00372-f008:**
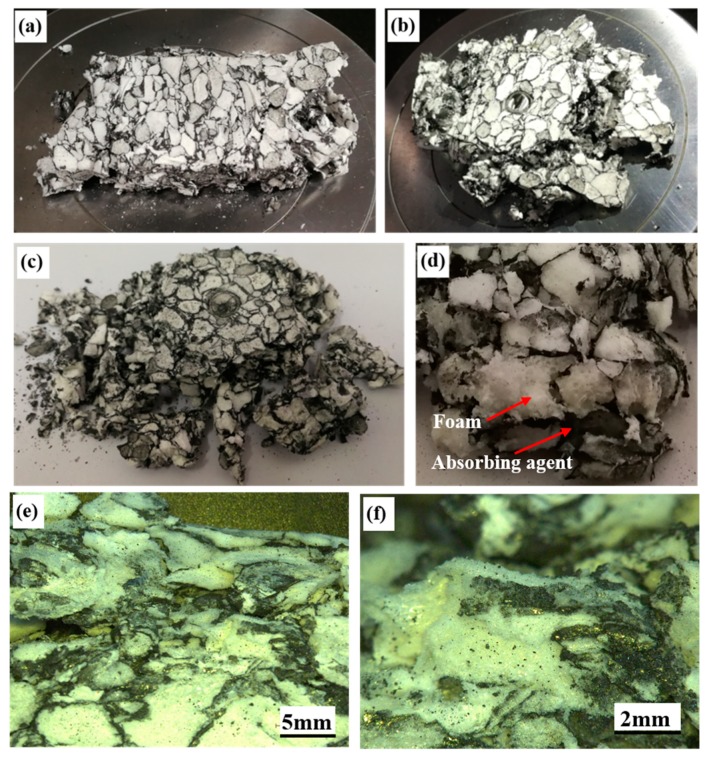
Absorbent PMI foam-based specimen images after compression. (**a**) Absorbent PMI foam; (**b**) Tube-enhanced absorbent PMI foam; (**c**) Foam-filled tube-enhanced absorbent PMI foam; (**d**) Fractography of the absorbent PMI foam showing interfaces between PMI foam and absorbing agent; (**e**,**f**) Local fracture surfaces of the absorbent PMI foam.

**Figure 9 polymers-11-00372-f009:**
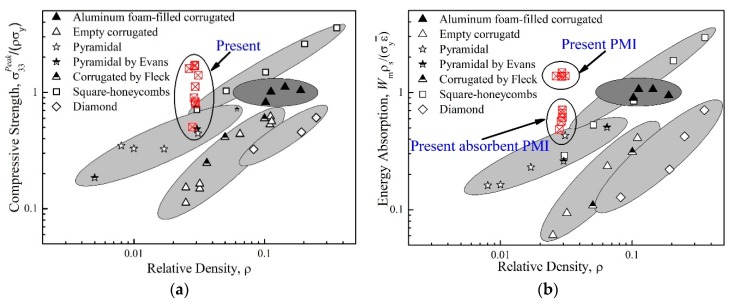
Comparison of present tube-enhanced normal and absorbent PMI foams with competing sandwich core designs [36]. (**a**) Specific compressive strength and (**b**) Specific energy absorption.

**Table 1 polymers-11-00372-t001:** Parameters of specimens for the compression tests; Al is 6061 aluminum alloy, steel is 304 stainless steel, Filled-Al is foam-filled aluminum tube, Filled-Steel is foam-filled 304 stainless steel tube.

Specimen	Foam Type	Tube Type	Averaged Mass (g)	Tube	Specimen Size
1	PMI Foam	None	13.46	None	40 × 40 × 40 mm^3^
2		Al	14.81	0.5 × Φ 10 mm	40 × 40 × 40 mm^3^
3		Filled-Al	14.88	0.5 × Φ 10 mm	40 × 40 × 40 mm^3^
4		Steel	15.79	0.2 × Φ 10 mm	40 × 40 × 40 mm^3^
5	Absorbent PMI foam	None	14.21	None	40 × 40 × 40 mm^3^
6		Al	14.99	0.5 × Φ 10 mm	40 × 40 × 40 mm^3^
7		Filled-Al	15.04	0.5 × Φ 10 mm	40 × 40 × 40 mm^3^
8		Steel	14.67	0.2 × Φ 10 mm	40 × 40 × 40 mm^3^
9		Filled-Steel	14.96	0.2 × Φ 10 mm	40 × 40 × 40 mm^3^

**Table 2 polymers-11-00372-t002:** Summary of the averaged elastic modulus E, compressive strength $\sigma_{\mathrm{peak}}$, and energy absorption per unit volume $W_{v}$ and per unit mass $W_{m}$ of the specimens as well as density $\rho_{c}$.

Specimen	$\rho_{c}\;({\mathrm{Kg}/m}^{3})$	*E*	$\sigma_{\mathrm{peak}}\;(\mathrm{MPa})$	$W_{v}\;(10^{3}\;\mathrm{KJ}/m^{3})$	$W_{m}\;(\mathrm{KJ}/\mathrm{Kg})$
1	210.3	318.6	8.90	3.84	18.26
2	231.4	386.0	10.31	4.20	18.15
3	246.7	566.4	9.14	4.53	18.36
4	232.5	354.9	10.53	4.53	19.48
5	222.0	231.7	2.96	1.42	6.40
6	234.2	338.0	5.14	2.07	8.84
7	235.0$\rho_{c}$	373.5	4.94	1.91	8.13
8	229.2	479.8	5.44	1.72	7.50
9	233.8	525.7	6.92	2.19	9.37

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
