# Peer review of "Enhancement by Metallic Tube Filling of the Mechanical Properties of Electromagnetic Wave Absorbent Polymethacrylimide Foam"

_polymers, 2019, doi:10.3390/polym11020372_

Round 1

Reviewer 1 Report

The manuscript entitled “Enhancement of metallic tube filling on mechanical properties of electromagnetic wave absorbent polyetherimide foam” uses metallic tube as a material adding in electromagnetic waves absorbent PMI foams to improve the mechanical property, which is logically organized. But there are some problems such as scientific writing specification, the mechanism is not sufficient, etc. I recommend major revisions. Detailed comments are as follows.

1.     On page 2 line 80, please add a space bar after the number and weight unit, check the similar problems and make corresponding revisions.

2.      Authors are suggested to provide EMI shielding absorption and reflection informationrespectively. Further compare the prepared sample with other studies.

3.     On page 5 line 122-126, please make sure the expression of " electromagnetic wave absorption efficiency formula " is correct.

4.     On page 5 line 130 please make sure the expression of “At the frequency of 5.2~7.3Hz” is correct. Besides, authors are suggested to provide shielding effectiveness (SSE) and absolute effectiveness (SSE/t) of the samples.

5.     Table 1, name of the specimens should be clearly defined and kept consistent.

6.     In section 3.2.1, the platform stresses should be unified at the same strain state.

7.     In Figure 3(a), please verify the purpose of the “curve circle positions”. In addition, why is the strain at the circle positions in Figure 3 (a) - (b) that it is not correspond with each other? Please elaborate in the results and discussion.

8.  The discussion of the energy absorption capacity (Wv and Wm) part of Section 3.2.2 is suggested to be mutually corroborated with the contents of Section 3.2.1. In addition, please provide mechanism to verify the schematic in Figure 3(b).

9.  In Figure 4, the thickness of the two samples is consistent, or not? What is the exact number of compression strain? Please provide scientific illustrations.

10.  In Section 3.2.3, please mark the crack form in the picture, in addition, authors are suggested to provide references to explain mechanism of “crack form”.

11. On page 6 line 155, authors are suggested to provide references to explain mechanism of “progressive buckling deformation”.

12. On page 9 line 223, figure 9 is not mentioned in the manuscript, please add.

Reviewer 2 Report

The authors report enhanced mechanical properties including elastic modulus, compressive strength and energy absorption by filling of metallic tubes into the pre-perforated holes of PMI foam blocks. The combination of enhanced mechanical properties and electromagnetic absorbing properties makes the tube-enhanced absorbent PMI foam quite competitive.  I recommend the manuscript for publication after some adjustments have been taken into account.

1) What is the morphology of metallic tubes? It is believed that the size and geometric structure plays an important role in determining the final mechanical properties.

2) The fracture surface should be characterized with high magnification for better discussion and understanding the mechanism.

3) The reviewer cannot find figure 9, the comparison figure. Please check and add it.

Round 2

Reviewer 2 Report

It can be accepted for publication.

Author Response

Dear Reviewer,

Thank you very much for all your efforts on our manuscript, which are much important for improving the quality of this manuscript entitled “Enhancement of metallic tube filling on mechanical properties of electromagnetic wave absorbent polyetherimide foam”. We have carefully checked the whole manuscript once again, especially for its English editing.

       Thank you very much for your acceptance for publication of the manuscript.

Yours sincerely,

L. L. Yan, Ph.D.

School of Aeronautics, Northwestern Polytechnical University

E-mail: yanleilei@nwpu.edu.cn

B. Han, Ph.D.

School of Mechanical Engineering, Xi'an Jiaotong University

E-mail: hanbinghost@qq.com